# Structural Characterization and Anti-Inflammatory Activity of a Novel Polysaccharide from *Duhaldea nervosa*

**DOI:** 10.3390/polym15092081

**Published:** 2023-04-27

**Authors:** Ziming Wang, Xueqin Ma, Silin Shi, Shuo He, Jian Li, Gidion Wilson, Wei Cai, Lianghong Liu

**Affiliations:** 1School of Pharmacy, Ningxia Medical University, Yinchuan 750004, China; 2Hunan Province Key Laboratory for Antibody-Based Drug and Intelligent Delivery System, School of Pharmaceutical Sciences, Hunan University of Medicine, Huaihua 418000, China

**Keywords:** *Duhaldea nervosa*, polysaccharide, structural characterization, anti-inflammatory activity, roots, isolation, RAW 264.7 cells, DNP−1

## Abstract

In the present study, a novel water-soluble polysaccharide (DNP−1) was isolated and purified from the root of *Duhaldea nervosa* via column chromatography. Structural analyses indicated that DNP−1 had a linear backbone consisting of (2→1)-linked β-D- fructofuranosyl residues, ending with a (2→1) bonded α-D-glucopyranose. DNP−1 was a homogeneous polysaccharide with an average molecular weight of 3.7 kDa. Furthermore, the anti-inflammatory activity of DNP−1 was investigated in vitro. The concentration of pro-inflammatory cytokines, including NO, TNF-α, MCP-1, IL-2, and IL-6, in the DNP−1 treatment group was suppressed in LPS-induced RAW 264.7 cells. DNP−1 was able to improve inflammatory injury by inhibiting the secretion of pro-inflammatory cytokines. These investigations into this polysaccharide from the root of *Duhaldea nervosa* provide a scientific basis for the further development of this plant. The results indicate that this *Duhaldea nervosa* polysaccharide could be used as a potential natural source for the treatment of inflammatory injury.

## 1. Introduction

Inflammation is a protective biological response of human tissues to pathogens, irritants, and other harmful stimuli [1]. However, insufficient or excessive inflammation increases the incidence of many diseases [2]. Inflammatory cells and mediators play an important role in the inflammatory response. Macrophages, which can regulate the immune system by producing various inflammatory and chemotactic factors, are especially important and have been widely used in the study of different inflammatory diseases. Nevertheless, previous studies have indicated that the long-term use of anti-inflammatory drugs could cause afflictive gastrointestinal reactions, including coagulopathy and liver injury [3,4]. Traditional Chinese medicines (TCM) have been widely used against inflammatory diseases owing to their more limited side effects [5].

*Duhaldea nervosa* (Wallich ex Candolle) A. Anderberg (*D. nervosa*), also known as “xiaoheiyao” and “maoxiucai” and belonging to the plant family Asteraceae (Compositae), is widely distributed in the southwestern provinces of China [6]. According to folk wisdom, especially that of the Dong minority, its roots could be used as medicine for treating fractures and bone wounds through external use or oral administration [7]. However, little attention has been devoted to the polysaccharides and the associated bioactivities of *D. nervosa* roots. Previous studies revealed the different pharmacological activities of extracts of *D. nervosa*, such as antioxidant and anti-inflammatory effects [8,9], which are associated with its bioactive compounds, such as steroids, terpenoids, and flavones [10,11], as well as chlorogenic acid analogues [12,13] and polysaccharides [14]. Previous studies indicated that *D. nervosa* has good anti-inflammatory activity [15]. However, to the best of our knowledge there are few studies on the polysaccharides from *D. nervosa.*

Polysaccharides are important biological macromolecules that exist universally in plants, fungi, and micro-organisms. Moreover, most of the natural polysaccharides from TCM are relatively safe for humans and animals, with low toxicity for living organisms [9]. In recent years, the use of polysaccharides as novel potential medicines or functional foods has drawn widespread attention because of their significant positive properties, including anti-inflammatory [16,17,18], antitumor [19], antioxidant [20], antidiabetes [21], and antivirus effects [22]. Most natural polysaccharides are biocompatible, biodegradable, and non-toxic. Therefore, polysaccharides have been widely used in health foods and other fields of human life. *D. nervosa* polysaccharides display intense mineralization effects and obvious mineralization nodules, which have a significant impact on quantitative evaluation [8].

Therefore, in present research a novel homogeneous polysaccharide (DNP−1) was extracted and purified from *D. nervosa* roots and its structure was identified. The in vitro anti-inflammatory activities of DNP−1 were also investigated. Structural characterization and anti-inflammatory activities might provide worthy guidance for in-depth research on polysaccharides from *D. nervosa*, laying the foundation for the comprehensive development of functional products in the future and providing a material basis for the clinical application of *D. nervosa*.

## 2. Materials and Methods

### 2.1. Materials and Reagents

A sample of *D. nervosa* roots was purchased from the Yunyao company in Yunnan, China. The plant used in this project was identified by Ye Wang, a Professor at the Hunan University of Medicine, as the root of *D. nervosa.* The voucher specimens (No. 2019112801) were deposited at the School of Pharmaceutical Sciences, Hunan University of Medicine. DEAE-52 cellulose and Sephadex G-200 were obtained from the Beijing Solarbio Science and Technology Co., Ltd. (Beijing, China). The Amicon Ultra centrifugation filter was produced by Millipore (Boston, MA, USA). Standard monosaccharides (fucose, galactosamine hydrochloride, glucose, rhamnose, glucosamine hydrochloride, N-acetylglucosamine, ribose, fructose, arabinose, xylose, mannose, galactose, glucuronic acid, mannuronic acid, galacturonic acid, and guluronic acid), lipopolysaccharide (LPS), and series dextrans (molecular weight 5.0 × 10^3^, 1.16 × 10^4^, 2.38 × 10^4^, 4.86 × 10^4^, 8.89 × 10^4^, 1.48 × 10^5^, and 2.73 × 10^5^ Da) were purchased from Sigma-Aldrich Co., Ltd. (St. Louis, MO, USA). The dextran of molecular weight at 1.15 × 10^3^ Da was purchased from Shanghaiyuanye Bio-Technology Co., Ltd. (Shanghai, China). Penicillin–streptomycin solution and Dulbecco’s Modified Eagle’s Medium (DMEM) were purchased from Thermo Fisher Scientific (Gibco, Waltham, MA, USA). Dexamethasone (DEX) was purchased from Shanghai Yien Chemical Technology Co., Ltd. (Shanghai, China). ELISA kits for IL-2, IL-6, TNFα, and MCP-1 were purchased from Affymetrix eBioscience (San Diego, CA, USA). The Nitric Oxide Assay Kit was purchased from Beyotime (Shanghai, China). All other reagents used were of analytical grade.

### 2.2. Extraction and Purification of Polysaccharides from the D. nervosa Roots

The dried powder of *D. nervosa* roots was pretreated with 95% (*v*/*v*) ethanol solution for 1 h in an 80 °C water bath to eliminate small lipophilic molecules and impurities. The hot water reflux extraction method [23] (solid–liquid ratio 1:20 (*w*/*v*), 90 °C, 2.5 h) was then used to extract the polysaccharides, with some modifications. The extract was centrifuged (at 20 min for 4000 r) to obtain a supernatant. The supernatant was evaporated in rotary evaporator at 50 °C under reduced pressure. Proteins were removed using the Sevag method [24] and the resulting liquid was fractionally precipitated with ethanol by adjusting the ethanol concentration to 20%, 40%, 70%, and 80%, respectively, for 24 h at 4 °C, as previously reported in [25], with some modifications. The precipitate was collected via centrifugation and lyophilized to obtain the crude polysaccharides DNP20, DNP40, DNP70, and DNP80. The heaviest crude polysaccharide—DNP70—will also be used for further separation and other fractions will be studied in the future.

The crude polysaccharide DNP70 (100 mg) was fully dissolved in distilled water (10 mL), placed in a DEAE-52 cellulose chromatography column (Φ3.0 × 150 cm), and eluted with 0, 0.05, 0.15, 0.3, and 0.5 mol/L NaCl solutions at a flow rate of 2.5 mL/min and monitored using the phenol–sulfuric acid assay. The DNP70-fr.1 was further purified through a Sephadex G-200 column (Φ3.5 × 180cm) and eluted with de-ionized water at a flow rate of 1 mL/min. Only one peak was obtained and filtered using a 10 kDa Amicon Ultra-15 centrifugal filter and centrifuged for 30 min at 5000× *g*. The centrifuge effluent was lyophilized for further research and named DNP−1.

### 2.3. Molecular Weight and Homogeneity Determination

The molecular weight and homogeneity of DNP−1 was measured using high-performance gel permeation chromatography (HPGPC) [26] with a Shimadzu LC-10A HPLC system equipped with a Shimadzu RI-10A refractive index detector and a BRT105-104-102 tandem gel column (8 × 300 mm, BoRui Saccharide Biotech Co. Ltd., Yangzhou, China). The mobile phase comprised 0.05M sodium chloride solution, which flowed at a rate of 0.6 mL/min. The DNP−1 solution (5.0 mg/mL) was then injected with a total elution time of 60 min in each run; the injection volume was 20 μL. The column was maintained at a temperature of 40 °C. The molecular weight calibration curve was determined using dextran standards. The molecular weight of DNP−1 was estimated through reference to the calibration curve created previously.

### 2.4. Monosaccharide Composition Analysis

High-performance anion exchange chromatography (HPAEC) [23] coupled with an electrochemical detector was used to determine the monosaccharide composition. DNP−1 (10 mg) was added into 10 mL of trifluoroacetic acid (TFA, 3M) and hydrolyzed for 3 h at 120 °C. The hydrolysate was blow-dried with nitrogen and dissolved with ultrapure water, centrifuged for 5 min at 12,000 r, and injected into the detection system (Dionex, ICS-5000, Thermo Fisher Scientific, USA) equipped with a CarbopacTMPA20 column (3 × 150 mm, Dionex). The elution program was as follows: 0.8% of 15 mM NaOH for the first 40 min, followed by the 0.8% of 15 mMNaOH with 100 mM NaOAc for the next 40 min. The flow rate was 1.0 mL/min.

### 2.5. Methylation Analysis

DNP−1 (7 mg) was dried overnight at 60 °C in a vacuum oven and dissolved in 5 mL DMSO and DMSO/NaOH solution (80 mg/mL, 500 μL) via sonication for 60 min. Iodomethane solution (500 μL) was added and reacted for 40 min in an ice-water bath with continuous stirring. Thereafter, 2 mL water was added to terminate the reaction.

The methylated polysaccharide was added into 2 mL TFA and hydrolyzed for 30 min at 50 °C; it was then dried via rotary evaporator at 40 °C and dissolved in 2 mL water. NaBH_4_ (20 mg) was added and reacted for 30 min at 40 °C. Acetic acid was added and dried with nitrogen. Amounts of 2 mL acetic anhydride and 2mL pyridine were added and reacted for 1 h at 95 °C. Methanol was added and dried via rotary evaporator, repeated four times. Amounts of 2.5 mL water and 2.5 mL trichloromethane were added and mixed well through vortexing. The trichloromethane phase was analyzed through gas chromatography/mass (GC/MS) (GCMS-7890A/5975C, Agilent, Santa Clara, CA, USA) with an HP-5MS capillary column (initial temperature, 80 °C; increased to 280 °C at 5 °C/min; held for 1 min at 280 °C).

### 2.6. Fourier-Transform Infrared (FT-IR) and UV–Visible (UV–Vis) Spectroscopic Scanning Analysis

The IR spectra of the DNP−1 was obtained using a Fourier transform infrared spectrometer (IR Affinity-1s, Shimadzu, Kyoto, Japan) in the frequency range of 400–4000 cm^−1^ [27]. A total of 1 mg of DNP−1 was mixed with 100 mg KBr powder, pressed into 1 mm pellets, and scanned for the FT-IR measurement.

The UV–vis spectra of DNP−1 was recorded using a UV–vis spectrometer (Unical Instrument Co., Ltd., Shanghai, China) in the spectral scanning range of 200–800 nm^−1^.

### 2.7. Nuclear Magnetic Resonance Spectroscopy Analysis

The structure details of DNP−1 was further investigated using NMR spectroscopy. The ^1^H NMR, ^13^C NMR, and 2D NMR (including 1H/1HCOSY, DEPT-135, HSQC, and HMBC) spectra of the DNP−1 sample (50 mg) were collected on a Bruker spectrometer (600 MHz) after deuterium was exchanged three times with D_2_O (0.5 mL, 99.9%), taking deuterated acetone as the internal reference.

### 2.8. Cell Culture and Viability Assay

The macrophage-like cell line RAW264.7 was purchased from the Cell Bank of Chinese Academy of Sciences (Shanghai, China). RAW264.7 cells were cultured in DMEM medium containing 10% (*v*/*v*) FBS, 100 µg/mL streptomycin, and 100 U/mL penicillin at 37 °C in 5% CO_2_ incubator.

The cell viability and cytotoxicity of DNP−1 on RAW264.7 cells were determined via MTT assay [28]. The RAW264.7 cells were adjusted and cultured in 96-well plates at a concentration of 1 × 10^5^ cells/well for 24 h and treated with different concentrations of DNP−1 (12.5, 25, 50, 100, and 150 µg/mL) for another 24 h at 37 °C. An equal volume of DMEM medium was used as a blank control. A total of 10 µL of MTT (5 mg/mL) was then added and incubated for 4 h at 37 °C. Subsequently, the MTT medium was removed, 100 µL of DMSO was added into every cell-hole to dissolve the formazan crystals and left for a few minutes at room temperature to ensure that all crystals were dissolved, and the absorbance was measured at 570 nm using the microplate reader (SPECTRO star Nano, BMG Labtech, Offenburg, Germany).

### 2.9. Detection of the Levels of Inflammatory Biomarkers

The enzyme-linked immunosorbent assay (ELISA) kits were used to determine the levels of inflammatory biomarkers [29]. RAW264.7 cells were seeded in 48-well plates at a density of 2 × 10^5^ cells per well for 24 h at 37 °C in a 5% CO_2_ incubator. The cells were pre-treated with DNP−1 (50, 100, 150 μg/mL) for 1 h and stimulated with lipopolysaccharide (LPS, 100 ng/mL) for 18 h. Cell supernatants were then collected and the levels of NO, TNF-α, MCP-1, IL-2, and IL-6 were determined by using ELISA kits following the manufacturer’s instructions. Dexamethasone (0.5 μM) was used as positive control.

### 2.10. Statistical Analysis

Data were shown as the means ± SE (bars) from at least three replicates for each assay. Statistical analysis was performed using GraphPad Prism software (GraphPad Prism version 9, San Diego, CA, USA) following a one-way analysis of variance (ANOVA). Results were considered as statistically significant when *p* < 0.05.

## 3. Results

### 3.1. Extraction and Purification of Polysaccharides from the D. nervosa Roots

The crude polysaccharide was extracted from *D. nervosa* roots and fractionally precipitated with ethanol. The crude polysaccharide DNP70 was purified using a DEAE-52 cellulose anion-exchange chromatography column and eluted with distilled water and 0.05, 0.15, 0.3, and 0.5 mol/L NaCl solutions (Figure 1A). Five separated peaks were acquired. This research was focused on DNP70-fr.1. The DNP70-fr.1 was further purified through Sephadex G-200 column chromatography and eluted with de-ionized water. The result showed a single peak of DNP−1 (Figure 1B), indicating that DNP−1 was a relatively homogeneous polysaccharide. Following filtration and lyophilization, DNP−1 powders were obtained.

The UV–vis spectra (Figure 1C) of the DNP−1 showed no absorption peaks at wavelengths of 260 and 280 nm in UV scanning spectrum within 200–800 nm, suggesting that DNP−1 was absent of nucleic acids and proteins.

### 3.2. Molecular Weight and Monosaccharide Composition Analysis

In this study, the average molecular weight and homogeneity of DNP−1 was analyzed via HPGPC. A symmetrical sharp and single peak was shown in Figure 1D, indicating that DNP−1 was a homogeneous polysaccharide. The standard regression equation was Y = −0.2028 X + 12.709, R^2^ = 0.9929, where Y represents the logarithm of molecular weight (lgMw) and X represents retention time (min). According to the standard regression equation of dextran standards and the elution time of the DNP−1, the molecular weight of DNP−1 was calculated to be 3.7 kDa.

The monosaccharide composition of DNP−1 was determined using an ion chromatography instrument and identified by comparing the retention time with standards (Figure 2A). Judging from the peak area, the DNP−1 mainly consisted of glucose and fructose at a molar ratio of 0.16:0.84 (Figure 2B).

### 3.3. Methylation Analysis 

Methylation analysis was used to elucidate the glycosidic linkage patterns. To acquire further structural information, the methylated products of DNP−1 were analyzed via GC-MS (Table 1 refer to Figure 2C). The GC-MS results demonstrated that DNP−1 has three types of sugar linkages: Fruf-(2→; Glcp-(1→; and →1)-Fruf-(2→, at a molar percentage ratio of 2.1:4.72:28.70, according to the peak areas. As fructose is a ketose, it isomerizes into mannoside and glucoside in the furan ring during the reduction process, as shown by the presence of the derivative 1,3,4,6-Me_4_ Manf/Glcf and 3,4,6-Me_3_-Manf/Glcf in methylation data. The methylation analysis was focused on qualitative rather than quantitative results because it is difficult to obtain standards for each individual monosaccharide derivative [30]. This issue could be the reason that the molar ratio of fructose and glucose in the methylation result was inconsistent with the monosaccharide composition analyses.

### 3.4. FT-IR Spectrum and NMR Analysis

FT-IR spectra (Figure 2D) were used to analyze the molecular properties of DNP−1. The absorption peak at 3361 cm^−1^ can be attributed to the -OH stretching of DNP−1. The weaker absorption peak at 2931cm^−1^ was attributed to C-H stretching vibration. The strong absorption peak at 1687 cm^−1^ was caused by the C=O and C=C stretching vibrations. The peak at 1422 cm^−1^ was caused by the −COOH stretching vibration. The three absorption peaks in the 950–1200 cm^−1^ range suggested that the DNP−1 had a pyranose ring. The absorption peaks at 934, 872, and 817 cm^−1^ indicated the presence of a fructofuranoside with β-bond types [31,32,33]. The FT-IR results were consistent with the monosaccharide composition analyses.

The structural characteristics of DNP−1 were further analyzed via one- and two-dimensional spectra, including ^1^H- NMR (Figure 3A), ^1^H-^1^H COSY (Figure 3B), ^13^C-NMR (Figure 3C), HSQC (Figure 3D), DEPT 135 (Figure 3E), and HMBC spectra (Figure 3F). The chemical shifts of DNP−1 mainly occurred in the region of 3.5 to 5.4 ppm and 60 to 110 ppm in the ^1^H NMR and ^13^C NMR spectra, respectively. These were typical signals for polysaccharide [34,35]. In the ^1^H NMR spectrum, the signals in the range of δ 5.0–5.9 ppm and δ 4.0–5.0 ppm indicated that DNP−1 contained α-glycosidical and β-glycosidical configurations [36]. The strong chemical shift at δ 4.70 ppm was the solvent of D_2_O. Comparing the chemical shifts with those in previous reports [33,37,38], the signal at 5.35 ppm (>5.0 ppm) was attributed to the anomeric protons of -α-D-Glcp-1→. Other hydrogen atom signals were attributed to the ^1^H-^1^H COSY and HSQC spectra. In the ^13^C spectrum, the signals at 104.61 and 93.97 ppm were attributed to C-2 of β fructosyl residues and C-1 of α-D-glucopyranosyl residues, respectively. The presence of β-D-fructofuranosyl linkage was confirmed using the signals at 82.41 ppm (C-5) on the DEPT 135 spectrum [39]. The ^13^C spectrum, DEPT-135 spectrum, and HSQC spectrum demonstrated that the signals at 72.60, 73.99, 70.60, 75.65, and 61.68 ppm could be attributed to the C-2, C-3, C-4, C-5, and C-6 of α-D-glucopyranosyl, respectively. The signals at 61.78, 78.30, 75.63, 82.41, and 63.88 ppm were attributed to the C-1, C-3, C-4, C-5, and C-6 of β-D-fructofuranosyl-2,1, respectively. These signals were entirely consistent with the signals outlined in the literature [40,41]. The chemical shifts in sugar residues are summarized in Table 2. The sequence of glycosidic residues in the DNP−1 was recorded using the HMBC spectrum. In the Figure 3F, the cross-signals at δ 5.35/104.61 ppm (H1/C2) revealed the existence of α-D-Glcp-1→2-β-D-Fruf-1→. In addition, other cross-signals at δ 3.85,3.65/104.61 ppm (H1a, H1b/C2), δ 4.18/104.61 ppm (H3/C2), δ 4.18/75.63 ppm (H3/C4), δ 4.01/78.30 ppm (H4/C3), δ 4.01/82.41 ppm (H4/C5), δ 4.01/63.88 ppm (H4/C6), δ 3.80/75.63 ppm (H5/C4), and δ 3.80/63.88 ppm (H5/C6) were determined as →2-β-D-Fruf-1→2-β-D-Fruf-1→. In addition, the peak area ratio of the terminal hydrogen of Glcp to fructose H-3 or H-4 was 1:22, while the estimated molecular weight of DNP−1 was 3744 Da, which was consistent with the previously determined results. The probable structure of DNP−1 was confirmed as α-D-Glcp-1→(2-β-D-Fruf-1)*n*→(*n* = 22) based on methylation analysis, chemical compositions, FT-IR spectrum analysis, and NMR spectra analysis. This result demonstrates that the polysaccharides extracted from *D. nervosa* were inulin-type fructans [37].

### 3.5. Effects of DNP−1 on RAW 264.7 Cell Viability

As shown in Figure 4A, the results indicated that DNP−1 (25, 50, 100, 150, and 200 µg/mL) displayed no significant toxicity toward RAW264.7 cells compared with the control.

### 3.6. Effects of DNP−1 on LPS-Induced Pro-Inflammatory Cytokine Production in RAW264.7 Cells

To evaluate whether DNP−1 affected the levels of pro-inflammatory cytokines, the secretions of NO, TNF-α, MCP-1, IL-2, and IL-6 in RAW264.7 cells were analyzed [29]. As shown in Figure 4B–F, RAW264.7 cells were treated with DNP−1 and pro-inflammatory cytokines were induced by LPS. Compared with the Control Group, all pro-inflammatory cytokines in the LPS-induced Group increased significantly (*p* < 0.001), indicating that the inflammation model was valid. Compared with the LPS-induced Group, the production of these pro-inflammatory cytokines was suppressed in the DNP−1 Treatment Group, in which the high-dose treatment was more effective than the low-dose treatment. DNP−1 treatment (150 μg/mL) resulted in low concentrations of NO, TNF-α, MCP-1, IL-2, and IL-6 in RAW 264.7 cells at levels that were significantly lower than those of the LPS-induced Group. These results demonstrated that DNP−1 was able to ameliorate inflammation by inhibiting the production of several pro-inflammatory cytokines at effective dosages.

## 4. Discussion

Studying the biological activity of TCM polysaccharides will help to broaden the sources of new drug developments. Although there are many studies on the biological activity of polysaccharides, structural analysis of polysaccharides is very difficult due to the large molecular weight of polysaccharides, the variety of glycosidic residues, and the existence of various modifying groups. Therefore, the structural properties and activities of the isolated *D. nervosa* homogeneous polysaccharides were characterized via various technical methods in this study.

In this report, a novel purified homogeneous polysaccharide DNP−1, with an average molecular weight of 3.7 kDa, was obtained from *D. nervosa*. Monosaccharide composition analysis indicated that the DNP−1 mainly consisted of glucose and fructose in a molar ratio of 0.16:0.84. This is inconsistent with the result that the peak area ratio of the terminal hydrogen of Glcp to fructose H-3 or H-4 was 1:22; this result may be due to the decomposition of fructose under acidic conditions [42,43,44], the fact that fructose can be transferred to mannitol and glucitol under reduction conditions [40,45], or the fact that the hydrolysis of the glycosidic bonds of fructo-oligosaccharides under high temperatures and strong acids results in glucose and fructose conjugates but not fructose [46]. IR, methylation, and NMR results revealed that the probable structure of DNP−1 was confirmed as α-D-Glcp-1 → (2-β-D-Fruf-1)*n*→(*n* = 22). However, in actual fact methylation analysis demonstrated that DNP−1 has three types of sugar linkages: Fruf-(2→; Glcp-(1→; and →1)-Fruf-(2→, at a molar percentage ratio of 2.1:4.72:28.70, according to the peak areas. As mentioned above, the difference in the monosaccharide molar ratio may be due to the reduction reaction of fructose and the limitations of the methylation analysis itself [32].

The biological functions of polysaccharides are closely related to their chemical compositions and structural characteristics, such as molecular mass, monosaccharide composition, and glycosidic bond type [47]. Many natural active polysaccharides are able to express anti-inflammatory activities by regulating the secretion of inflammatory factors, signaling pathways, the immune system, etc. [48]. As the body’s first line of defense in the removal of foreign bodies, macrophages are distributed in various tissues and organs [49] and play a key role in the human immune response and the phagocytosis of pathogens [50]. LPS induces the differentiation of macrophages into M1 macrophages, which are typically characterized by the production of pro-inflammatory cytokines [51]. The release of large amounts of pro-inflammatory factors, such as TNF-α, IL-2, MCP-1, IL-6, and NO, induces the secretion of adhesion molecules from vascular endothelial cells, leading to the aggregation of neutrophils, monocytes, and lymphocytes. Eventually, these cells migrate to the surface of the injured tissue and cause tissue necrosis [52]. Anti-inflammatory levels were significantly reduced in macrophages in the DNP−1 group in this study. DNP−1 in the concentration range of 12.5–150 μg/mL showed no cytotoxic effects; in addition, DNP−1 attenuated the LPS-induced secretion of TNF-α, IL-2, MCP-1, IL-6, and NO in RAW264.7 macrophages, showing good anti-inflammatory potential. The results of this paper can be further used to study the anti-inflammatory mechanism of *D. nervosa* and suggest that *D. nervosa* polysaccharide could be developed as a natural anti-inflammatory material in the future. However, further research is needed to elucidate the relationship between the precise mechanism and structure activity.

## 5. Conclusions

In this study, a novel polysaccharide (DNP−1) was purified from the crude polysaccharide DNP70 of *D. nervosa* through DEAE-52 cellulose and Sephadex G-200 column chromatography. The molecular weight and monosaccharide composition showed that DNP−1 was composed of glucose and fructose with a molecular weight of 3.7 kDa. The chemical structure of DNP−1 was measured via FT-IR, GC/MS, and NMR. The probable structure of DNP−1 was α-D-Glcp-1→(2-β-D-Fruf-1) *n*→. Moreover, the anti-inflammatory activities of DNP−1 were investigated in vitro. The pro-inflammatory cytokine production in RAW264.7 cells could be decreased by DNP−1, which suggests that DNP−1 has the potential to be used as an immunomodulator in food and pharmaceutical industries. Moreover, this study can be utilized as a research basis for understanding the relationship between the polysaccharide structure and anti-inflammatory activity, thus providing a material basis for the clinical application of *D. nervosa*.

## Figures and Tables

**Figure 1 polymers-15-02081-f001:**
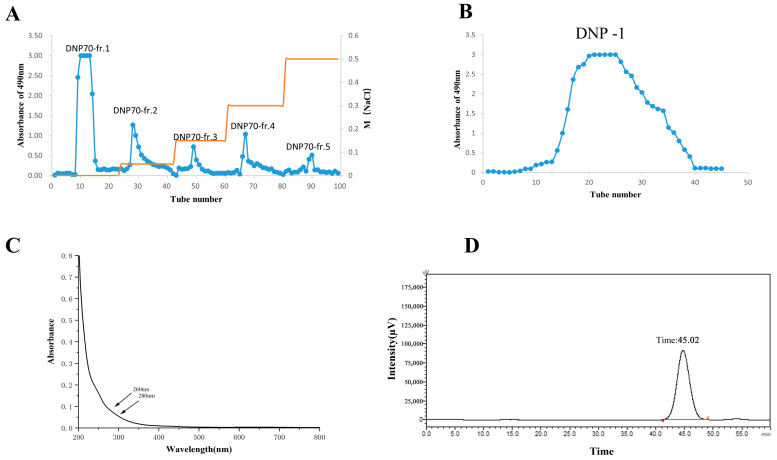
(**A**) Eluted profile DNP70 on DEAE-cellulose column; (**B**) eluted profile of DNP70-fr.1 on Sephadex G-200 column; (**C**) UV–vis spectrum of DNP−1; and (**D**) HPGPC analysis of DNP−1.

**Figure 2 polymers-15-02081-f002:**
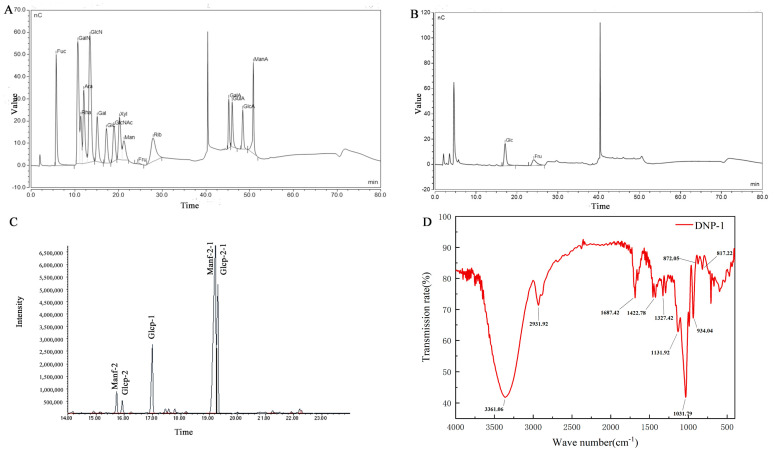
(**A**) HPIEC of monosaccharide standards; (**B**) HPIEC of DNP−1; (**C**) methylation and GC-MS chromatogram of DNP−1; and (**D**) FT-IR spectra of DNP−1.

**Figure 3 polymers-15-02081-f003:**
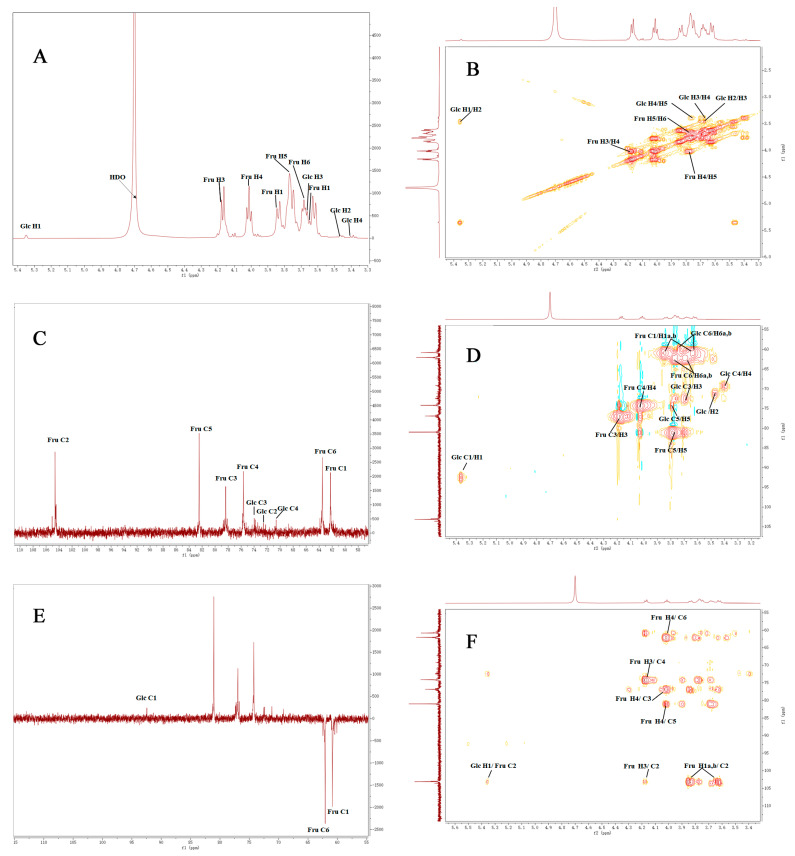
(**A**) ^1^H NMR spectrum of DNP−1 in D_2_O; (**B**) ^1^H-^1^H COSY NMR spectrum of DNP−1 in D_2_O; (**C**) ^13^C NMR spectrum of DNP−1 in D_2_O; (**D**) HSQC NMR spectrum of DNP−1 in D_2_O; (**E**) DEPT-135 NMR spectrum of DNP−1 in D_2_O; and (**F**) HMBC NMR spectrum of DNP−1 in D_2_O.

**Figure 4 polymers-15-02081-f004:**
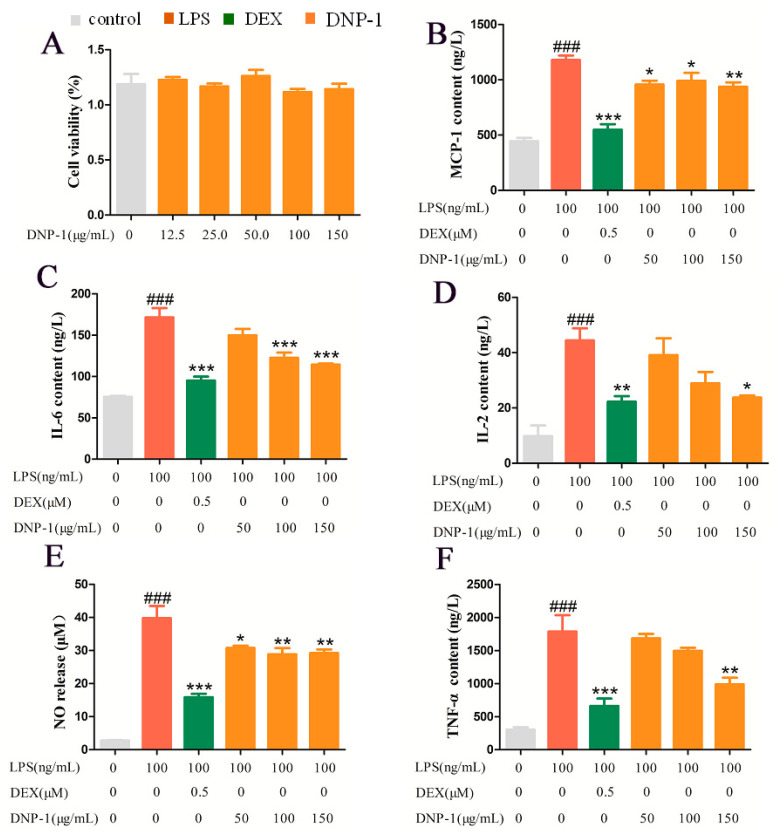
Effect of DNP−1 on cell viability of RAW 264.7 cells (**A**). Effects of DNP−1 on expression levels of MCP-1 (**B**), IL-6 (**C**), IL-2 (**D**), and TNF-α (**F**) in RAW 264.7 cells induced by LPS (100 ng/mL). Effect of DNP−1 on production levels of NO in RAW 264.7 cells induced by LPS (100 ng/mL) (**E**). Data values are expressed as mean ± SE (bars) (*n* ≥ 3). Significant difference: * *p* < 0.05, ** *p* < 0.01, and *** (*p* < 0.001) for difference from the LPS Group. ### (*p* < 0.001) for difference from Control Group.

**Table 1 polymers-15-02081-t001:** Methylation analysis of DNP−1 by using GC-MS.

Methylated Sugar	Mass Fragments (*m*/*z*)	Molar Ratios (%)	Deduced Linkage
1,3,4,6-Me_4_ Manf/Glcf	87, 101, 129, 145, 161	2.1	Fruf-(2→
2,3,4,6-Me_4_-Glcp	43, 71, 87, 101, 117, 129, 145, 161, 205	4.72	Glcp-(1→
3,4,6-Me_3_-Manf/Glcf	43, 71, 87, 99, 101, 129, 145, 161, 189	28.70	→1)-Fruf-(2→

**Table 2 polymers-15-02081-t002:** Assignments of ^1^H and ^13^C NMR spectra of DNP−1 based on analysis of COSY, HSQC, and HMBC spectra.

Glycosyl Residues	C1/H1a, b	C2/H2	C3/H3	C4/H4	C5/H5	C6/H6a, b
α-D-Glcp-1	93.97/5.35	72.60/3.47	73.99/3.68	70.60/3.40	75.65/3.79	61.68/3.72,3.62
β-D-Fruf-2,1	61.78/3.85,3.65	104.61/ns	78.30/4.18	75.63/4.01	82.41/3.80	63.88/3.68,3.76

## Data Availability

The data presented in this study are available on request from the corresponding author.

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
