# Peer review of "Structural Characterization and Anti-Inflammatory Activity of a Novel Polysaccharide from Duhaldea nervosa"

_polymers, 2023, doi:10.3390/polym15092081_

Round 1
Reviewer 1 Report
The manuscript “Structural characterization and anti-inflammatory activity of a novel polysaccharide from Duhaldea nervosa” requires improvement. Authors should revise and improve the English language and grammar throughout the manuscript. English proofreading is strongly recommended to enhance the overall presentation of the manuscript.
In general, the manuscript is not following to the format of the Journal. Inconsistencies in font, size and etc. were detected throughout the manuscript. Scientific name of the plant should be in italic. The resolution of the Figures was low and were placed not in line with its appearance in text. Authors should provide high resolution Figures. Reference style is not consistent.
Introduction was not well outlined and it is rather brief. It should start from a general subject to a specific subject of the topic. Lack of focus and significance in this section. Plant’s name was repeated in the first sentence. Local name should use “…” to indicate in a sentence. Rewritten is strongly needed for this section.
Materials and Method: Plant was not authenticated by a botanist with voucher specimen. Why heaviest crude polysaccharides DNP70 is selected? How about others? Some experimental was not thoroughly described, including the full equipment name, model and manufacturer.
In Results and Discussion, what is the significance of isolating DNP-1? What is the purpose of isolating DNP-1 from the plant? Any previous studies on the similar genus? Knowledge gap? Include a Table for 1H and 13C of DNP-1 and its literature values and cite the reference.
In conclusion, it should include the significant findings from the study, not merely repeating the results.
Author Response
Dear reviewer,
Thank you for your handling my manuscript. Your comments were highly insightful and enabled us to greatly improve the quality of the manuscript. In the following pages are our point-by-point responses to each of the comments.
Comments and Suggestions for Authors
Point 1: The manuscript “Structural characterization and anti-inflammatory activity of a novel polysaccharide from Duhaldea nervosa” requires improvement. Authors should revise and improve the English language and grammar throughout the manuscript. English proofreading is strongly recommended to enhance the overall presentation of the manuscript.
Response 1: Thank you for your valuable and thoughtful comments. As review’s comment, the language was polished by https://www.mdpi.com/authors/english.
Point 2: In general, the manuscript is not following to the format of the Journal. Inconsistencies in font, size and etc. were detected throughout the manuscript. Scientific name of the plant should be in italic. The resolution of the Figures was low and were placed not in line with its appearance in text. Authors should provide high resolution Figures. Reference style is not consistent.
Response 2: Thank you for your valuable and thoughtful comments. As review’s comment, improvements have been made to font, size, image resolution, and other issues. Reference style was revised according the reviewer’s suggestion.
Point 3: Introduction was not well outlined and it is rather brief. It should start from a general subject to a specific subject of the topic. Lack of focus and significance in this section. Plant’s name was repeated in the first sentence. Local name should use “…” to indicate in a sentence. Rewritten is strongly needed for this section.
Response 3: Thank you for your valuable and thoughtful comments. It was revised according the reviewer’s suggestion.
Point 4: Materials and Method: Plant was not authenticated by a botanist with voucher specimen. Why heaviest crude polysaccharides DNP70 is selected? How about others? Some experimental was not thoroughly described, including the full equipment name, model and manufacturer.
Response 4: Thank you for your valuable and thoughtful comments. First, it has been added in materials and methods. All Duhaldea nervosa samples used in this project was identified by Ye Wang (professor, Hunan University of Medicine), as the root of the plant Duhaldea nervosa. The voucher specimens were deposited in the School of Pharmaceutical Sciences, Hunan University of Medicine. In addition, due to the highest yield of polysaccharides, DNP-70 is prioritized for further separation and purification in this study, and other components will be investigated in subsequent studies. Finally, some experimental was the whole described, including the full equipment name, model and manufacturer. It was revised according the reviewer’s suggestion.
Point 5: In Results and Discussion, what is the significance of isolating DNP-1? What is the purpose of isolating DNP-1 from the plant? Any previous studies on the similar genus? Knowledge gap? Include a Table for 1H and 13C of DNP-1 and its literature values and cite the reference.
Response 5: Thank you for your valuable and thoughtful comments. DNP-1 is the first homogeneous polysaccharide whose structure and anti-inflammatory activities has been analyzed in this study.. The purpose of isolating DNP-1 from plants is to obtain a novel homogeneous polysaccharide from Duhaldea nervosa. Meanwhile, understanding the structure–function relationship of DNP-1 can provide material basis for Duhaldea nervosa’s clinical application and lay the foundation for the comprehensive development of this plant’s functional products in the future. And previous studies have shown that Duhaldea nervosa is a good source of of polysaccharides[1], and Duhaldea nervosa has anti-inflammatory activity[2]. But, at present, there is no research on polysaccharides of similar species and genera. Among them the literature values of 1H- NMR and 13C-NMR of DNP-1 is cited in the arcicle[3-5].
- Liu, L.H.; Wang, L.T.; Zhi, X.J.; Wang, X.G.; Jiang, R.F.; Tan, Z.L. Preliminary Test of Chemical Components and Content Determination of Total Polysaccharides in Dong Medicine Duhaldea nervosa. China Pharmaceuticals 2020, 29 (9) , 96-99. https://doi.org/10.3969/j.issn.1006-4931.2020.09.029
- Ning,Q.; Jiang, Y.J.; Zhu, J.H.; Zhou, L.; Cai, W.; Tang, N.Y. Preparation of Chitosan-based Whitening and Anti-inflammatory Films Using Angelica Dahurica Benth and Duhaldea Nervosa. SHANDONG CHEMICAL INDUSTRY 2019, 48, 14-16. https://doi.org/10.19319/j.cnki.issn.1008-021x.2019.23.006
- Pontes, A. G.; Silva, K. L.; Fonseca, S. G.; Soares, A. A.; Feitosa, J. P.; Braz-Filho, R.; Romero, N. R.; Bandeira, M. A., Identification and determination of the inulin content in the roots of the Northeast Brazilian species Pombalia calceolaria L. Carbohydr Polym 2016, 149, 391-8. https://doi.org/10.1016/j.carbpol.2016.04.108
- Meng, Y.; Xu, Y.; Chang, C.; Qiu, Z.; Hu, J.; Wu, Y.; Zhang, B.; Zheng, G., Extraction, characterization and anti-inflammatory activities of an inulin-type fructan from Codonopsis pilosula. Int J Biol Macromol 2020, 163, 1677-1686. https://doi.org/10.1016/j.ijbiomac.2020.09.117
- Shao, X.; Sun, C.; Tang, X.; Zhang, X.; Han, D.; Liang, S.; Qu, R.; Hui, X.; Shan, Y.; Hu, L.; Fang, H.; Zhang, H.; Wu, X.; Chen, C., Anti-Inflammatory and Intestinal Microbiota Modulation Properties of Jinxiang Garlic (Allium sativum L.) Polysaccharides toward Dextran Sodium Sulfate-Induced Colitis. J Agric Food Chem 2020, 68, (44), 12295-12309. https://doi.org/10.1021/acs.jafc.0c04773
Point 6: In conclusion, it should include the significant findings from the study, not merely repeating the results.
Response 6: Thank you for your valuable and thoughtful comments. It was revised according the reviewer’s suggestion.

Reviewer 2 Report
The manuscript entitled 'Structural characterization and anti-inflammatory activity of a novel polysaccharide from Duhaldea nervosa' needs a minor revision. It contains a significant amount of experimental research conducted in laboratory conditions (in vitro).
The main notes:
1. The moderate check is required for the English language and style through the text, for instance: Abstract: In the present study, a novel water-soluble polysaccharide (DNP-1) was isolated and purified from the root of Duhaldea nervosa via column chromatography. Structural analyses indicated that DNP-1 was a linear backbone consisteding of (2→1)-linked β-D- fructofuranosyl residues, ending with a (2→1) bonded α-D-glucopyranose. DNP-1 was a homogeneous polysaccharide with an average molecular weight of 3.7 kDa. Furthermore, the anti-inflammatory activity of DNP-1 was also investigated. The concentration of pro-inflammatory cytokines including NO, TNF-α, MCP-1, IL-2, and IL-6 in the DNP-1 treatment group were was suppressed in LPS-induced RAW 264.7 cells. DNP-1 could improve the inflammatory injury by inhibiting the secretion of pro-inflammatory cytokines. These investigations of the polysaccharide from the root of Duhaldea nervosa provide a scientific basis for further development of this plant of Duhaldea nervosa. The results indicated that the Duhaldea nervosa polysaccharide could be used as a potential natural source in treating an inflammatory injury.
2. The list of keywords could be expanded to capture the main findings. For instance, the words 'roots' and 'isolation' could be added.
3. Regarding 'Introduction'.
Why the Latin name of this plant is mentioned twice in this 1st line? Duhaldea nervosa (Wallich ex Candolle) A. Anderberg (Duhaldea nervosa). Maybe the first one could be replaced with its common name ' Veined-Leaf Inula'?
It should be added the newer name Asteraceae - 'to the plants' family Asteraceae (Compositae)'.
4. It needs to improve the objectives and rationale of the study. So, it is worth processing and adding at least 5 new sources.
5. The purpose of the study should be redone as it is like an Abstract.
6. The list of sources does not list any scientific publications for 2022 or 2023. I propose adding several new sources to improve the background.
7. It should be noted in the Abstract, Methods and Conclusions that the anti-inflammatory activity was studied 'in vitro'.
8. The italic type should be used everywhere for writing Latin names of genera and species, for instance, Duhaldea nervosa - in 2.1. Materials and reagents and 2.2.
9. Figures 1-3 are not clear enough. Please, correct them.
Author Response
Dear reviewer,
Thank you for your handling my manuscript. Your comments were highly insightful and enabled us to greatly improve the quality of the manuscript. In the following pages are our point-by-point responses to each of the comments.
Comments and Suggestions for Authors
The manuscript entitled 'Structural characterization and anti-inflammatory activity of a novel polysaccharide from Duhaldea nervosa' needs a minor revision. It contains a significant amount of experimental research conducted in laboratory conditions (in vitro).
The main notes:
Point 1:The moderate check is required for the English language and style through the text, for instance: Abstract: In the present study, a novel water-soluble polysaccharide (DNP-1) was isolated and purified from the root of Duhaldea nervosa via column chromatography. Structural analyses indicated that DNP-1 was a linear backbone consisteding of (2→1)-linked β-D- fructofuranosyl residues, ending with a (2→1) bonded α-D-glucopyranose. DNP-1 was a homogeneous polysaccharide with an average molecular weight of 3.7 kDa. Furthermore, the anti-inflammatory activity of DNP-1 was also investigated. The concentration of pro-inflammatory cytokines including NO, TNF-α, MCP-1, IL-2, and IL-6 in the DNP-1 treatment group were was suppressed in LPS-induced RAW 264.7 cells. DNP-1 could improve the inflammatory injury by inhibiting the secretion of pro-inflammatory cytokines. These investigations of the polysaccharide from the root of Duhaldea nervosa provide a scientific basis for further development of this plant of Duhaldea nervosa. The results indicated that the Duhaldea nervosa polysaccharide could be used as a potential natural source in treating an inflammatory injury.
Response 1: Thank you for your valuable and thoughtful comments. The language has been polished by https://www.mdpi.com/authors/english.
Point 2: The list of keywords could be expanded to capture the main findings. For instance, the words 'roots' and 'isolation' could be added.
Response 2: Thank you for your valuable and thoughtful comments. It was revised according the reviewer’s suggestion.
Point 3: Regarding 'Introduction'.
Why the Latin name of this plant is mentioned twice in this 1st line? Duhaldea nervosa (Wallich ex Candolle) A. Anderberg (Duhaldea nervosa). Maybe the first one could be replaced with its common name ' Veined-Leaf Inula'?
It should be added the newer name Asteraceae - 'to the plants' family Asteraceae (Compositae)'.
Response 3: Thank you for your valuable and thoughtful comments. .It was revised according the reviewer’s suggestion. Moreover, our previous study has is pointed out that Duhaldea nervosa’s Latin name had been revised, the older name: Inula nervosa Wall, is no longer used. Its latest Latin scientific name is Duhaldea nervosa (Wallich ex Candolle) A. Anderberg[1].
- Guan, Y.; Wang, Y.; Zhou, Y.; Wang, Y.W.; Zheng, B.J.; Wang, L.T.; Cai W. Determination of Isochlorogenic acid A and Isochlorogenic acid C in Duhaldea nervosa by HPLC. LISHIZHEN MEDICINE AND MATERIA MEDICA RESEARCH 2017, 28, 1032-1034. https://doi.org/10.3969/j.issn.1008-0805.2017.05.004
Point 4: It needs to improve the objectives and rationale of the study. So, it is worth processing and adding at least 5 new sources.
Response 4: Thank you for your valuable and thoughtful comments. It was revised according the reviewer’s suggestion.
Point 5: The purpose of the study should be redone as it is like an Abstract.
Response 5: Thank you for your valuable and thoughtful comments. It was revised according the reviewer’s suggestion.
Point 6: The list of sources does not list any scientific publications for 2022 or 2023. I propose adding several new sources to improve the background.
Response 6: Thank you for your valuable and thoughtful comments. It was revised according the reviewer’s suggestion.
Point 7: It should be noted in the Abstract, Methods and Conclusions that the anti-inflammatory activity was studied 'in vitro'.
Response 7: Thank you for your valuable and thoughtful comments. It was revised according the reviewer’s suggestion.
Point 8: The italic type should be used everywhere for writing Latin names of genera and species, for instance, Duhaldea nervosa - in 2.1. Materials and reagents and 2.2.
Response 8: Thank you for your valuable and thoughtful comments. It was revised according the reviewer’s suggestion.
Point 9: Figures 1-3 are not clear enough. Please, correct them.
Response 9: Thank you for your valuable and thoughtful comments. It was revised according the reviewer’s suggestion.

Round 2
Reviewer 1 Report
D. nervosa should be in italic. Please thoroughly revise throughout the manuscript.
It should be “xiaoheiyao”.
Include the voucher specimen number in 2.1
Author Response
Dear reviewer,
Thank you for your handling my manuscript. Your comments were highly insightful and enabled us to greatly improve the quality of the manuscript. In the following pages are our point-by-point responses to each of the comments.
Comments and Suggestions for Authors
Point 1: D. nervosa should be in italic. Please thoroughly revise throughout the manuscript.
Response 1: Thank you for your valuable and thoughtful comments. “D. nervosa” has been italicized throughout the manuscript.
Point 2: It should be “xiaoheiyao”.
Response 2: Thank you for your valuable and thoughtful comments. It was revised according the reviewer’s suggestion.
Point 3: Include the voucher specimen number in 2.1
Response 3: Thank you for your valuable and thoughtful comments. The voucher specimen number: No.2019112801 has been added to 2.1
